# Factors associated with visual outcomes after cataract surgery: A cross-sectional or retrospective study in Liberia

**Rohit C. Khanna**[1,2,3,4], **Varsha M. Rathi**[1,2]*, **Edward Guizie**[5], **Gurcharan Singh**[1,5], **Kumar Nishant**[1,5], **Smrita Sandhu**[1,5], **Rajashekar Varda**[1,2], **Anthony Vipin Das**[2,6], **Gullapalli Nageswara Rao**[1,2]

1 Allen Foster Community Eye Health Research Centre, Gullapalli Pratibha Rao International Centre for Advancement of Rural Eyecare, LV Prasad Eye Institute, Hyderabad, India, 2 Brien Holden Eye Research Centre, LV Prasad Eye Institute, Hyderabad, India, 3 School of Optometry and Vision Science, University of New South Wales, Sydney, Australia, 4 University of Rochester, School of Medicine and Dentistry, Rochester, NY, United States of America, 5 Liberia Eye Centre, John F. Kennedy Memorial Medical Centre, Monrovia, Liberia, 6 Department of eyeSmart EMR & AEye, L V Prasad Eye Institute, Hyderabad, India

* varsha@lvpei.org

**Data Availability Statement:** All relevant data are within the manuscript and its Supporting Information files.

## Abstract

### Objective

To report the initial outcomes and associated risk factors for poor outcome of cataract surgery performed in Liberia

### Methods and analysis

LV Prasad Eye Institute (LVPEI), Hyderabad, started providing eye care in Liberia since July 2017. Electronic Medical Records of 573 patients operated for age-related cataract from July 2017 to January 2019 were reviewed. One eye per patient was included for analysis. All patients underwent either phacoemulsification or manual small incision cataract surgery (MSICS). Pre and postoperative uncorrected visual acuity (UCVA) and best-corrected visual acuity (BCVA) were recorded at one day, 1–3 weeks and 4–11 weeks. Main outcome measure was BCVA at 4–11 weeks; Intraoperative complications and preoperative ocular comorbidities (POC) were noted. BCVA less than 6/12 was classified as visual impairment (VI). Risk factor for VI was analysed using the logistic regression model.

### Results

Of the 573 patients, 288 were males and 285 were females (49.7%). Mean age was 65.9 ±10.9 years; 14.3% had POC. The surgical technique was mainly MSICS (94.59%, n = 542). At 4–11 weeks, good outcome of 6/12 or better was noted in 38.55% (UCVA) and 82.54% (BCVA). Visual acuity (VA) of 6/18 or better as UCVA and BCVA was noted in 63.5% and 88% eyes respectively. Poor outcome of less than 6/60 was noted as UCVA (11.11%) and BCVA (5.22%). Multivariable analysis showed poor visual outcomes significantly higher in patients with POC (odds ratio 3.28; 95% CI: 1.70, 6.34).

**Funding:** The authors received no specific funding for this work.

**Competing interests:** The authors have declared that no competing interests exist.

## Conclusion

The cataract surgical outcomes in Liberia were good; with ocular comorbidities as the only risk factor.

## Introduction

There are 253 million people blind or visually impaired worldwide. Of these 89% live in low and middle income countries (LMIC); and 55% are women. [1] Globally, cataract is the leading cause of blindness and second leading cause of visual impairment (VI). [2] In Sub-Saharan Africa 35–45% of blindness and 25–35% of VI is caused by cataract. [2] Liberia in West Africa has a population of approximately 5 million. [3] World Health Organization (WHO) estimates that approximately 1% of Liberians (approximately 35,000 people) suffer from blindness, of which 50% is due to cataract. [3] In terms of human resources, the sub-Saharan region has on average only one ophthalmologist per one million population. [4] This holds true even for Liberia. [4] The Cataract Surgical Rate (CSR) is one of the lowest in Liberia with 81 in 2010 and 157 in 2014. [5, 6] In terms of cataract surgical outcomes, a study by Frucht-Pery and Feldman on cataract surgery in patients with leprosy in Liberia reported that a visual acuity of 20/200 or better was achieved only in 65% of patients. [7] However, this study included only 43 eyes of 30 patients. [7] There are similar reports from other African countries with poor outcomes ranging from 10–40%. [8, 9]

The L V Prasad Eye Institute's (LVPEI) pyramidal model of eye care delivery has a Centre of Excellence (CoE) at the top, catering to a population of 50 million population, with Tertiary Centres (TC) at the next level, each for 5 million population. [10] These are linked to Secondary Centres (SC) covering 0.5–1 million population mostly in rural locations, with Vision Centres (VC) at primary level for 50,000 population, and Vision Guardians (VG) for 5,000 population. The functions at each level of the pyramid are clearly delineated and demarcated. The SCs are run by one or two ophthalmologists who are trained at a TC or COE for a year. Patients from SCs are referred to TCs or COE only for advanced care and management of complex problems. [10] The model has shown good outcomes of cataract surgery in their rural SCs. [11] The secondary centres provide comprehensive affordable, accessible and appropriate eye care irrespective of the paying capacity of the patients, and have fully equipped surgical facilities where mainly cataract surgeries and other procedures are performed. [10]

In 2014, LVPEI launched the Liberia Eye Health Initiative (LEHI), at the John F Kennedy (JFK) Medical Centre, which is the apex centre for health care in Liberia. A formal collaborative agreement was signed between LVPEI and JFK. The Liberia Eye Centre (LEC) was formally inaugurated on 24 July 2017 in Monrovia to provide comprehensive eye care for people of Liberia. All the staff in the LEC were trained at LVPEI. Since 2018, LVPEI also started the ophthalmology residency training program in Liberia. The purpose of this study is to evaluate outcomes of cataract surgery performed at the LEC as well as associated risk factors for poor outcomes.

## Materials and methods

We retrospectively analysed the records of patients who underwent cataract surgery in the LEC at JFK Memorial Medical Centre, Monrovia, Liberia from July 2017 to January 2019. The study was approved by the University of Liberia-Pacific Institute for Research & Evaluation

Institution Review Board. The study was conducted in accordance to the tenets of Declaration of Helsinki.

The study included all patients 40 years of age or above. Only the first operated eye was included in the analysis. Data was collected from the Electronic Medical Records (EMR). The protocols were similar to described in our previous publication. [11] In brief, the patients underwent comprehensive eye examination, which included detailed history; uncorrected visual acuity (UCVA) and best corrected visual acuity (BCVA); intraocular pressure measurement with Goldmann applanation tonometer; slit lamp examination; dilated lens examination to assess the lens status; and stereoscopic fundus examination with +78/90 Dioptre lens as well as indirect ophthalmoscope. In case there was no view of retina, a B-scan ultrasound was done to rule out any posterior segment pathology. Pre-existing ocular comorbidities were grouped as corneal pathologies, retinal disease, glaucoma and others (non-glaucomatous optic nerve pathologies and uveitis). The systemic comorbidities included hypertension (HT), diabetes mellitus (DM), and HIV seropositive patients. When patient was advised surgery, protocols similar to L V Prasad Eye Institute protocols were followed. [12, 13] In brief, when patient was advised surgery, a designated counsellor did the counselling and explained the type of surgeries as well as associated risk and benefits. A day prior to surgery, intraocular lens (IOL) power calculation was done by measuring keratometry and A Scan biometry. Informed consent from patient and attendant was also taken along with routine blood pressure and blood sugar measurement and a physician fitness a day prior to surgery. On day of surgery, prior to entering operating room, patient dress was changed and eye were dilated with plain tropicamide (0.8% w/v) eye drop. Local anaesthesia given with 2% Xylocaine. After local anaesthesia, patient was shifted to operating room and under all aseptic precautions, eye was cleaned with betadine and draped. The surgeon decided on the surgical technique–either a phacoemulsification or manual small incision cataract surgery (MSICS) with or without intraocular lens (IOL) implantation. MSICS was performed by standard Blumenthal technique. [14] For phacoemulsification, a 5.5 mm scleral would was constructed and a routine phacoemulsification was performed. The choice of procedure was left to surgeon discretion. Anterior chamber (AC) and posterior chamber (PC) IOLs made up of polymethyl methacrylate (PMMA) were used. The intraoperative complications (posterior capsule rupture, zonular dehiscence etc) were noted. Posterior capsular rent or zonular dehiscence was managed by automated vitrectomy and placement of IOL was based on the available support of anterior and / or posterior capsule.

All the surgeries were performed by three surgeons as well as other visiting faculty. All three surgeons had experience of performing more than 1,500 cataract surgery. Post-operatively, patient was prescribed topical steroids for 4 weeks in tapering doses and topical antibiotics for a week.

The primary outcome measure was postoperative visual acuity. The visual acuity was noted preoperatively and postoperatively on day 1, between 1–3 weeks and 4–11 weeks of surgery and analysed. Visual acuity was categorized as 6/12 or better; less than 6/12 to 6/18; less than 6/18 to 6/60 and less than 6/60. Poor outcome was defined as BCVA of less than 6/12 in the operated eye.

## Statistical analysis

Statistical analysis was done using Stata 13 (Statacorp, Texas).

Logistic regression model was used to evaluate the association of risk factors for poor outcomes and Fisher's Exact test was used for categorical variable. A two tailed p value of <0.05 was considered statistically significant. Risk factors for poor outcomes were analysed using univariable and multivariable regressions, based on BCVA at 4–11 weeks follow up.

## Results

Between July 2017 and January 2019, a total of 739 cataract surgeries were performed. Of these, 126 were bilateral cataract surgeries (252 eyes), and their first operated eye was included in the analysis. Those 40 patients who underwent cataract surgery at age less than 40 years were also excluded. Hence, 573 eyes of 573 patients were included in the study. Table 1 shows the baseline demographics and ocular findings of these 573 patients. The mean age of the patients was 65.9±10.9 years (ranging from 40 to 99 years); and 49.7% were females. Pre-existing ocular comorbidity was present in 82 patients (14.3%) and systemic comorbidity was noted in 20 patients (3.5%).

Of the 573 eyes, the MSICS technique was used in 542 (94.6%) eyes and phacoemulsification was done in 31 (5.4%) eyes. Five hundred and sixty two (98.1%) eyes underwent PCIOL, 11 (1.9%) eyes were either left aphakic or had an ACIOL. Major intraoperative complications such as posterior capsular rest and zonular dehiscence were noted in 3.3% of eyes. Other complications (0.9%) such as Descemet's membrane detachment (one eye), iridodialysis (two eyes) and would leak (two eyes) were also noted.

Table 2 shows the preoperative and post-operative (uncorrected and best corrected) visual acuity on the first day; 1 to 3 weeks and 4–11 weeks follow up. At last follow up, UCVA of 6/12 or better was seen in 38.6% patients and BCVA of 6/12 or better was seen in 82.5% patients. Similarly, UCVA of 6/18 or better was seen in 63.5% patients and BCVA of 6/18 or better was

**Table 1. Demographics and baseline characteristics of the patients who underwent cataract surgery.**

| Variable | | Number of patients (Percentage) |
|---|---|---|
| **Age group** | 40 to 49 years | 38 (6.6) |
| | 50 to 59 years | 122 (21.3) |
| | 60 to 69 years | 202 (35.3) |
| | 70 years or above | 211 (36.8) |
| **Gender** | Male | 288 (50.3) |
| | Female | 285 (49.7) |
| **Economic status** | Paying (Paid surgeries) | 306 (53.4) |
| | Non-paying (Free surgeries) | 267 (46.6) |
| **Eye** | Right eye | 329 (57.4) |
| | Left eye | 244 (42.6) |
| **Preoperative BCVA*** | Less than 6/60 | 289 (50.4) |
| | Less than 6/18 to 6/60 | 94 (16.4) |
| | Less than 6/12 to 6/18 | 75 (13.1) |
| | 6/12 or better | 115 (20.1) |
| **Systemic comorbidities** | Diabetes Mellitus | 4 (0.7) |
| | Hypertension | 12 (2.1) |
| | HIV seropositive | 4 (0.7) |
| **Ocular comorbidities** | None | 491 (85.7) |
| | Cornea | 16 (2.8) |
| | Glaucoma | 39 (6.8) |
| | Retina | 9 (1.6) |
| | Others[#] | 18 (3.1) |

*BCVA Best corrected visual acuity

#Nonglaucomatous optic nerve disease and Uveitis

**Table 2. Uncorrected and best corrected visual acuity at different follow-up intervals.**

| | Preoperative | | Post-operative | | | | | |
| | | | Day 1 | | 1–3 weeks | | 4–11 weeks | |
| | n = 573 (100%) | | n = 573 (100%) | | n = 509 (88.83%) | | n = 441 (76.96%) | |
| VA | UCVA[#] | BCVA[*] | UCVA[#] | BCVA[*] | UCVA[#] | BCVA[*] | UCVA[#] | BCVA[*] |
| | n (%) | n (%) | n (%) | n (%) | n (%) | n (%) | n (%) | n (%) |
| 6/12 or better | 44 (7.7) | 115 (20.1) | 249 (43.5) | 419 (73.1) | 224 (44.0) | 402 (79.0) | 170 (38.5) | 364 (82.5) |
| Less than 6/12 to 6/18 | 48 (8.4) | 75 (13.1) | 150 (26.2) | 50 (8.7) | 118 (23.2) | 33 (6.5) | 110 (24.9) | 24 (5.4) |
| Less than 6/18 to 6/60 | 132 (23.0) | 94 (16.4) | 114 (19.9) | 53 (9.2) | 109 (21.4) | 35 (6.9) | 112 (25.4) | 30 (6.8) |
| Less than 6/60 | 349 (60.9) | 289 (50.4) | 60 (10.5) | 51 (8.9) | 58 (11.4) | 39 (7.7) | 49 (11.1) | 23 (5.2) |

[#]UCVA Uncorrected visual acuity

[*]BCVA Best corrected visual acuity

seen in 88% patients. UCVA and BCVA of less than 6/60 were seen in 11.11% and 5.22% patients respectively.

At 4–11 weeks, there were 441 (77%) patients available for follow-up. Table 3 shows the demographics and ocular findings of those who reported for follow up and those who did not, between 4–11 weeks. There was no difference between those lost to follow up and those who were available for follow up, in terms of gender, paying and non-paying status, eye operated, type of surgery, use of intraocular lens, choice on intracameral antibiotics, and intraoperative complications. However, there was a difference in terms of age group, operating surgeon and presence of ocular comorbidity.

Table 4 shows the differences between the demographics and ocular characteristics of patients who had good versus poor outcomes. Those with poor outcomes were older (p = 0.04); had higher intraoperative complications (p<0.001) as well as associated ocular comorbidity (<0.001); and had either an AC IOL or were left aphakic (p = 0.001).

Table 5 shows univariable and multivariable association of demographic and ocular surgical factors with poor outcomes (BCVA of less than 6/12 at 4–11 weeks follow-up). In univariable analysis, poor outcomes were associated with presence of ACIOL/No IOL (OR 6.07; 95% CI:1.80, 20.43); intraoperative complications (OR 5.10; 95% CI: 1.73, 15.00); and associated ocular comorbidities (OR 3.61; 95% CI: 2.05, 6.37). However in multivariable analysis, the only significant factor was presence of ocular comorbidities (OR 3.28; 95% CI: 1.70, 6.34).

## Discussion

One of the indicators for quality of cataract surgery is the outcome of surgery. Hence, periodic monitoring of outcomes should be an essential component of any hospital quality audit. We reported here the outcomes of cataract surgeries performed in a present facility built in Liberia. There is very little data on outcomes of cataract surgery from Liberia, due to limited number of surgeries. We started this new facility in July 2017, and installed the Electronic Medical Records (EMR) system to collect and store patient data on a daily basis. This is a good source for monitoring the outcomes of cataract surgery.

The only report from Liberia, by Frucht-Pery and Feldman, on cataract surgical outcomes in patients with leprosy, showed that a visual acuity of 20/200 or better was achieved in 65% of patients. [7] However, this study was published more than 25 years ago, included only 43 eyes and the surgical technique used was intracapsular or extracapsular cataract extraction. [7] In our setting, a majority of the surgeries were MSICS with IOL implantation, and had better outcomes. UCVA of 6/18 or better was seen in 63.5% patients and BCVA of 6/18 or better was

**Table 3. Demographics and ocular findings of those who followed up and who did not follow up between 4–11 weeks of follow up.**

| Variables | Sub-group | Available | Not available | P value |
|---|---|---|---|---|
| | | n = 441 | n = 132 | |
| **Age group** | 40–49 years | 34 (89.5%) | 4 (10.5%) | |
| | 50–59 years | 90 (73.8%) | 32 (26.2%) | |
| | 60–69 years | 146 (72.3%) | 56 (27.7%) | |
| | > = 70 years | 171 (81%) | 40 (19%) | |
| | | | | 0.04 |
| **Gender** | Male | 219 (76%) | 69 (24%) | |
| | Female | 222 (77.9%) | 63 (22.1%) | |
| | | | | 0.60 |
| **Economic status** | Paying(Paid surgeries) | 227 (74.2%) | 79 (25.8%) | |
| | Non-paying (Free surgeries) | 214 (80.1%) | 53 (19.9%) | |
| | | | | 0.09 |
| **Eye** | Right eye | 255 (77.5%) | 74 (22.5%) | |
| | Left eye | 186 (76.2%) | 58 (23.8%) | |
| | | | | 0.72 |
| **Type of surgery** | Phacoemulsification | 26 (83.9%) | 5 (16.1%) | |
| | MSICS# | 415 (76.6%) | 127 (23.4%) | |
| | | | | 0.35 |
| **Type of IOL** | PC IOL^ | 430 (76.5%) | 132 (23.5%) | |
| | No IOL* or AC IOL& | 11 (100%) | 0 (0.0%) | |
| | | | | 0.07 |
| **Intracameral antibiotics** | Cefuroxime | 49 (83.1%) | 10 (16.9%) | |
| | Moxifloxacin | 334 (74.1%) | 117 (25.9%) | |
| | | | | 0.21 |
| **Intraoperative Complications** | No or minor complications | 427 (77.1%) | 127 (22.9%) | |
| | Major complications | 14 (73.7%) | 5 (26.3%) | |
| | | | | 0.73 |
| **Surgeon category** | Faculty 1 | 140 (82.8%) | 29 (17.2%) | |
| | Faculty 2 | 249 (82.5%) | 53 (17.5%) | |
| | Faculty 3 | 46 (75.4%) | 15 (24.6%) | |
| | Visiting Faculty | 6 (14.6%) | 35 (85.4%) | |
| | | | | <0.001 |
| **Systemic comorbidities** | Absent | 425 (76.9%) | 128 (23.1%) | |
| | Present | 16 (80%) | 4 (20%) | |
| | | | | 0.74 |
| **Ocular comorbidities** | Absent | 370 (75.4%) | 121 (24.6%) | |
| | Present | 71 (86.6%) | 11 (13.4%) | |
| | | | | 0.03 |

#MSICS–Manual small incision cataract surgery

*IOL–Intraocular lens

^PCIOL–Posterior chamber intraocular lens

&AC IOL–Anterior chamber intraocular lens

seen in 88% patients at last follow up. With further cut-off in visual acuity value to 6/12 or better, UCVA of 6/12 or better was seen in 38.6% patients and BCVA of 6/12 or better was seen in 82.5% patient. Similar outcomes were reported in many other countries in Asia and India.

**Table 4. Difference between demographics and ocular characteristic of good versus poor outcome.**

| Variables | Sub-group | BCVA 6/12 or better | BCVA less than 6/12 | Total | P value |
|---|---|---|---|---|---|
| | | n = 364 | n = 77 | n = 441 | |
| **Age group** | 40–49 years | 29 (85.3%) | 5 (14.7%) | 34 | |
| | 50–59 years | 79 (87.8%) | 11 (12.2%) | 90 | |
| | 60–69 years | 126 (86.3%) | 20 (13.7%) | 146 | |
| | > = 70 years | 130 (76%) | 41 (24%) | 171 | |
| | | | | | 0.04 |
| **Gender** | Male | 178 (81.3%) | 41 (18.7%) | 219 | |
| | Female | 186 (83.8%) | 36 (16.2%) | 222 | |
| | | | | | 0.49 |
| **Economic status** | Paying(Paid surgeries) | 190 (83.7%) | 37 (16.3%) | 227 | |
| | Non-paying (Free surgeries) | 174 (81.3%) | 40 (18.7%) | 214 | |
| | | | | | 0.51 |
| **Eye** | Right eye | 207 (81.2%) | 48 (18.8%) | 255 | |
| | Left eye | 157 (84.4%) | 29 (15.6%) | 186 | |
| | | | | | 0.38 |
| **Type of surgery** | Phacoemulsification | 22 (84.6%) | 4 (15.4%) | 26 | |
| | MSICS# | 342 (82.4%) | 73 (17.6%) | 415 | |
| | | | | | 0.77 |
| **Type of IOL** | PC IOL^ | 359 (83.5%) | 71 (16.5%) | | |
| | No IOL* or AC IOL& | 5 (45.5%) | 6 (54.5%) | | |
| | | | | | 0.001 |
| **Intracameral antibiotics** | Cefuroxime | 36 (73.5%) | 13 (26.5%) | 49 | |
| | Moxifloxacin | 281 (84.1%) | 53 (15.9%) | 334 | |
| | | | | | 0.07 |
| **Intraoperative complications** | Absent | 357 (83.6%) | 70 (16.4%) | 427 | |
| | Present | 7 (50%) | 7 (50%) | 14 | |
| | | | | | <0.001 |
| **Surgeon category** | Faculty 1 | 119 (85%) | 21 (15%) | 140 | |
| | Faculty 2 | 197 (79.1%) | 52 (20.9%) | 249 | |
| | Faculty 3 | 43 (93.5%) | 3 (6.5%) | 46 | |
| | Visiting Faculty | 5 (83.3%) | 1 (16.7%) | 6 | |
| | | | | | 0.09 |
| **Systemic comorbidities** | Absent | 351 (82.6%) | 74 (17.4%) | 425 | |
| | Present | 13 (81.3%) | 3 (18.7%) | 16 | |
| | | | | | 0.89 |
| **Ocular comorbidities** | Absent | 319 (86.2%) | 51 (13.8%) | 370 | |
| | Present | 45 (63.4%) | 26 (36.6%) | 71 | |
| | | | | | <0.001 |

#MSICS–Manual small incision cataract surgery

*IOL–Intraocular lens

^PCIOL–Posterior chamber intraocular lens

&AC IOL–Anterior chamber intraocular lens

[15–18] MSICS was the primary surgical technique in all these studies also. [15–18] There are very few reports on outcomes of cataract surgery from Africa, and outcomes reported by some of these are not encouraging. [19–21] One of the reasons for improved outcome in some

**Table 5. Univariable and multivariable association of demographics, surgical factors with visual outcome of 6/12 or better at 4–11 weeks follow-up.**

| Variables | Univariable analysis | | Multivariable analysis | |
|---|---|---|---|---|
| | Odds Ratio (95% CI$^\$$) | P value | Odds Ratio (95% CI$^\$$) | P value |
| **Age group** | | | | |
| 40–49 years | Reference | | Reference | |
| 50–59 years | 0.81(0.26, 2.52) | 0.71 | 1.16 (0.27, 4.96) | 0.84 |
| 60–69 years | 0.92 (0.32, 2.66) | 0.88 | 1.29 (0.33, 5.06) | 0.71 |
| > = 70 years | 1.83 (0.66, 5.03) | 0.24 | 3.02 (0.81, 11.28) | 0.1 |
| **Gender** | | | | |
| Male | Reference | | Reference | |
| Female | 0.84 (0.51, 1.38) | 0.49 | 0.81(0.45, 1.47) | 0.49 |
| **Economic status** | | | | |
| Paying(Paid surgeries) | Reference | | Reference | |
| Non-paying (Free surgeries) | 1.18 (0.72, 1.93) | 0.51 | 1.00 (0.55, 1.81) | 0.99 |
| **Eye** | | | | |
| Right eye | Reference | | Reference | |
| Left eye | 0.80 (0.48, 1.32) | 0.38 | 0.82 (0.46, 1.49) | 0.52 |
| **Type of surgery** | | | | |
| Phacoemulsification | Reference | | Reference | |
| MSICS$^\#$ | 1.17 (0.39, 3.51) | 0.77 | 0.65 (0.19, 2.20) | 0.49 |
| **Type of IOL** | | | | |
| PC IOL^ | Reference | | Reference | |
| No IOL* or AC IOL$^\&$ | 6.07 (1.80, 20.43) | 0 | 3.97 (0.50, 31.66) | 0.19 |
| **Intracameral antibiotics** | | | | |
| Cefuroxime | Reference | | Reference | |
| Moxifloxacin | 0.52 (0.26, 1.05) | 0.07 | 0.59 (0.27, 1.30) | 0.19 |
| **Intraoperative Complications** | | | | |
| Absent | Reference | | Reference | |
| Present | 5.10 (1.73, 15.00) | <0.001 | 4.30 (0.68, 27.01) | 0.12 |
| **Surgeon category** | | | | |
| Faculty 1 | Reference | | Reference | |
| Faculty 2 | 1.50 (0.86, 2.61) | 0.16 | 1.67 (0.79, 3.54) | 0.18 |
| Faculty 3 | 0.40 (0.11, 1.39) | 0.15 | 0.68 (0.17, 2.73) | 0.59 |
| Visiting Faculty | 1.13 (0.13, 10.19) | 0.91 | 0.96 (0.09, 10.67) | 0.97 |
| **Systemic comorbidities** | | | | |
| Absent | Reference | | Reference | |
| Present | 1.09 (0.30, 3.94) | 0.89 | 0.93 (0.23, 3.81) | 0.92 |
| **Ocular comorbidities** | | | | |
| Absent | Reference | | Reference | |
| Present | 3.61 (2.05, 6.37) | <0.001 | 3.28 (1.70, 6.34) | <0.001 |

#MSICS–Manual small incision cataract surgery

*IOL–Intraocular lens

^PCIOL–Posterior chamber intraocular lens

&AC IOL–Anterior chamber intraocular lens

$CI–Confidence interval

centres may be the availability of well-trained surgeons performing high volume work in these centres. However, outcomes of this study was better than many other studies from Africa, including PRECOG study. [22–25] This could be due to availability of well-trained surgeons,

accurate biometry in all cases, availability of equipment and consumables, and well trained staff. [26] Another possible reason for poor outcomes reported in Africa could be because the earlier outcome studies in Africa were based on extracapsular cataract extraction (ECCE). In our study, surgeries were performed using MSICS technique. The outcomes of MSICS have been reported to be better than ECCE. [15–18, 22] WHO has recommended a good visual outcome as 90% having BCVA of 6/18 or better and 80% having UCVA of 6/18 or better. The outcomes were also similar to the recommended rates of WHO for BCVA, but were less than what is recommended for UCVA. [27] There were a significant number of patients blind or with severe visual impairment (SVI) (VA less than 6/60) preoperatively; whereas only 5% had VA of less than 6/60 postoperatively at 4–11 weeks. This implies better quality of life for those blind or with SVI before cataract surgery. This also suggests that there are a significant number of blind and SVI in the population who need intervention on a priority basis.

Major intraoperative complications were found in 3.3% eyes, and these are within acceptable standard stated by WHO (less than 5%). These were mainly due to posterior capsular rent or zonular dehiscence. This was slightly higher than other studies from India and Nepal [15–18] but less than most studies from Africa. [23–25] Mavrakanas et al reported an overall complication rate of 8.2% with less in MSICS surgeries (5.3%) as compared to ECCE (10.2%). [23] However, these surgeries were carried out by cataract surgeons undergoing training. [23] Complication rate was also much lower than what was reported on PRECOG and other studies. [25] However, complications were higher compared to Matta et al (1.4%) and Sherwin et al (2.5%). [11, 28] Unlike other studies, we did not find age or presence of a complication to be risk factor for poor outcomes. [20, 29] This could be due to lesser number of complications as well as adequate management of these complications. [29] No devastating complications like expulsive choroidal haemorrhage or endophthalmitis were reported, though the number of surgeries was less.

In univariable analysis, poor outcomes were associated with presence of ACIOL/No IOL (OR 6.07; 95% CI:1.80, 20.43), intraoperative complications (OR 5.10; 95% CI: 1.73, 15.00), and associated ocular comorbidities (OR 3.61; 95% CI: 2.05, 6.37). However in multivariable analysis, the only significant factor was presence of ocular comorbidities (OR3.28; 95% CI: 1.70, 6.34). This was similar to the reports from other studies. [15, 20, 22] This suggests that a good preoperative comprehensive eye examination by a trained ophthalmologist would be necessary to identify ocular comorbidities that are likely to affect outcomes. Patients with ocular comorbidities can then be counselled with a clear explanation on the outcomes of surgery.

One of the limitations of the study could be the patients who were lost to last follow up (132 patients, 23%). However, on comparing those available and those lost to follow up, there was a statistically significant difference between the two groups only in terms of age group, operating surgeon, and presence of ocular comorbidity. Those available for follow up had a higher prevalence of ocular comorbidity. Hence, this could underestimate the good outcomes at last follow up. PRECOG study showed good correlation between early outcome (3 or fewer days) and 40 days or more post operatively. It also showed that eyes with borderline or poor outcome at discharge tend to improve and achieve better vision at 4–6 months. Hence, the results obtained from those available for follow up may underestimate the good outcomes.

This is the first study to report on the outcomes of cataract surgeries in Liberia. We believe that there are multiple factors which might have played a role—availability of full time well-trained surgeons, accurate biometry, availability of equipment and consumables, and well trained staff. However, further studies are needed to support this hypothesis. A Residency program has been started in Liberia recently and this article is one of the first articles from Liberia and cataract surgery outcomes of residents will be compared and reported in future.

In conclusion, overall the outcomes of cataract surgery in Liberia was good as compared to many studies done in Africa. Apart from this, the complications rates were also comparable to WHO standards and only risk factor for poor outcome was presence of ocular comorbidities.

## Supporting information

**S1 Data. 'liberia0402019_age40_and_above.dta' attached.**
(DTA)

## Author Contributions

**Conceptualization:** Rohit C. Khanna, Varsha M. Rathi, Gullapalli Nageswara Rao.

**Data curation:** Rohit C. Khanna, Varsha M. Rathi, Anthony Vipin Das.

**Formal analysis:** Rohit C. Khanna, Varsha M. Rathi.

**Investigation:** Gurcharan Singh, Kumar Nishant, Smrita Sandhu, Rajashekar Varda.

**Methodology:** Rohit C. Khanna, Varsha M. Rathi, Edward Guizie, Gurcharan Singh, Kumar Nishant, Smrita Sandhu, Rajashekar Varda, Anthony Vipin Das, Gullapalli Nageswara Rao.

**Project administration:** Edward Guizie, Gurcharan Singh, Kumar Nishant, Smrita Sandhu, Rajashekar Varda.

**Resources:** Rohit C. Khanna, Varsha M. Rathi, Edward Guizie, Gurcharan Singh, Kumar Nishant, Smrita Sandhu, Rajashekar Varda, Anthony Vipin Das, Gullapalli Nageswara Rao.

**Software:** Rohit C. Khanna.

**Supervision:** Rohit C. Khanna, Edward Guizie, Gurcharan Singh, Kumar Nishant, Smrita Sandhu, Rajashekar Varda, Gullapalli Nageswara Rao.

**Validation:** Rohit C. Khanna, Anthony Vipin Das.

**Visualization:** Gullapalli Nageswara Rao.

**Writing – original draft:** Rohit C. Khanna, Varsha M. Rathi.

**Writing – review & editing:** Rohit C. Khanna, Varsha M. Rathi, Edward Guizie, Gurcharan Singh, Kumar Nishant, Smrita Sandhu, Rajashekar Varda, Anthony Vipin Das, Gullapalli Nageswara Rao.

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
