## [Decision Letter · Decision Letter 0]

3 Mar 2020

PONE-D-19-34050

Cataract Surgery Visual Outcomes and Associated Risk Factors in Liberia

PLOS ONE

Dear Professor Khanna,

Thank you for submitting your manuscript to PLOS ONE. After careful consideration, we feel that it has merit but does not fully meet PLOS ONE’s publication criteria as it currently stands. Therefore, we invite you to submit a revised version of the manuscript that addresses the points raised during the review process.

The comments are attached.

We would appreciate receiving your revised manuscript by Apr 17 2020 11:59PM. To enhance the reproducibility of your results, we recommend that if applicable you deposit your laboratory protocols in protocols.io, where a protocol can be assigned its own identifier (DOI) such that it can be cited independently in the future. For instructions see: http://journals.plos.org/plosone/s/submission-guidelines#loc-laboratory-protocols

We look forward to receiving your revised manuscript.

Kind regards,

Fakir M Amirul Islam, PhD

Academic Editor

PLOS ONE

Journal Requirements:

a)    Please provide an amended Funding Statement that declares *all* the funding or sources of support received during this specific study (whether external or internal to your organization) as detailed online in our guide for authors at http://journals.plos.org/plosone/s/submit-now.  

b)    Please state what role the funders took in the study.  If any authors received a salary from any of your funders, please state which authors and which funder. If the funders had no role, please state: "The funders had no role in study design, data collection and analysis, decision to publish, or preparation of the manuscript."

Reviewers' comments:

Reviewer's Responses to Questions

**Comments to the Author**

1. Is the manuscript technically sound, and do the data support the conclusions?

Reviewer #1: Yes

Reviewer #2: No

Reviewer #3: Yes

2. Has the statistical analysis been performed appropriately and rigorously? 

Reviewer #1: Yes

Reviewer #2: Yes

Reviewer #3: Yes

3. Have the authors made all data underlying the findings in their manuscript fully available?

Reviewer #1: No

Reviewer #2: Yes

Reviewer #3: Yes

4. Is the manuscript presented in an intelligible fashion and written in standard English?

Reviewer #1: Yes

Reviewer #2: No

Reviewer #3: Yes

5. Review Comments to the Author

Reviewer #1: This study presents the cataract surgery outcomes of the Liberia Eye Health Initiative by the LVPEI and highlights risk factors for poor outcome. The study is simple and the results clearly presented, but some grammar editing and proofreading is required to facilitate readability. Recalculation of some figures in the tables would also be helpful. I have included some suggestions below.

Line 28: ‘Since July 2017, LV’

Putting this at end of sentence would improve flow

Line 44: recommendation, not ‘recommended’

Line 51: of, not ‘for’

Line 54: from blindness of which 50% is due to cataract.

Add comma after blindness

Line 54-56: In terms of human resources, there is only one ophthalmologist for one million population on an average in the sub-Saharan region.

Rephrase:

In terms of human resources, the sub-Saharan region has on average only one ophthalmologist per one million population.

Line 56: This holds true even for Liberia [4]

Add full stop at end of sentence

Line 63-65 : The L V Prasad Eye Institute’s (LVPEI) pyramidal model of eye care delivery has a Centre of Excellence (CoE) at the top catering to a population of 50 million population with Tertiary Centres (TC) at the next level, each for 5 million population.[10]

add commas after 'top' and '50 million' to improve clarity

Line 80: The Liberia Eye Centre (LEC)

80 was formally inaugurated on 24 July 2017 in Monrovia, to provide comprehensive eye care

81 for people of Liberia.

Delete comma

Line 81-82

Since 2018, LVPEI also started the ophthalmology residency training program in Liberia.

Suggest delete this line: irrelevant and disrupts the flow (unless this study involves this program?)

Line 82-84

The purpose of this study is to evaluate outcomes of cataract surgery performed, at the LEC as well as associated risk factors for poor outcomes.

Delete comma

Line 132: While 562 (98.1%) eyes underwent PCIOL, 11 (1.9 %) eyes were either left aphakic or had an ACIOL.

Rephrase; try not to start sentence with 'while'

Line 133-136: Major intraoperative complications such as posterior capsular rest and zonular dehiscence were noted in 3.3% eyes. Other complications (0.9%) such as Descemet’s membrane detachment (1eye), iridodialysis (2 eyes) and would leak (2 eyes) were also noted.

-33% 'of' eyes

- Note spacing between ‘1’ and ‘eye’

- Generally, when quoting figures below ten, they should be spelt in full (eg. one eye)

Line 138-139: Table 2 shows the preoperative and post-operative (uncorrected and best corrected) visual acuity first day; 1 to 3 weeks and 4-11 weeks follow up.

…visual acuity ‘on the’ first day

Line 162: Those with poor outcomes were older (p=0.04); had higher intraoperative complications (p<0.001) as well as associated ocular comorbidity (<0.001); and had no IOL and AC IOL (p=0.001).

It would be clearer to say ‘and had either an ACIOL or were left aphakic’

Line 183: We reported here the outcomes of cataract surgeries performed by in a new facility

184 built in Liberia.

Replace with 'present'

Line 185: ‘We started this new facility in July 2017, and installed the Electronic Medical Records (EMR) system to collect and store patient data on a daily basis. This is a good source for monitoring the outcomes of cataract surgery.’

Comment: This part is not relevant in discussion. Please delete. Any discussion of methods eg EMR should be in Methods.

Line 189: The only report from Liberia by Frucht-Pery and Feldman, on cataract surgical outcomes in 190 patients with leprosy, showed that a visual acuity of 20/200 or better was achieved in 65% of 191 patients.

Add comma after Liberia

Line 191: However, this study published more than 25 years ago, included only 43 eyes and

192 the surgical technique used was intracapsular or extracapsular cataract extraction.

-grammar: ‘…this study, which was published ...’

Line 198-202: There are very few reports on outcomes of cataract surgery from Africa, and outcomes reported by these are not encouraging, except a few centres. One of the reasons for this was the availability of well-trained surgeons performing high volume work in these centres. However, outcomes were better than many other studies from Africa, including PRECOG study. This could be due to availability 203 of well-trained surgeons, accurate biometry in all cases, availability of equipment and

consumables, and well trained staff.

Please rephrase for clarity and readability. For example, by 'one of the reasons for this', it is unclear if you mean the reason for the poor outcome or the exceptions. and when you say' outcomes were better than many other studies', to which outcomes do you refer?

Line 208: The outcomes were also similar to the recommended rates of WHO for BCVA (BCVA of 90% having 6/18 or better), but were less than what is recommended for UCVA.

Rephrase sentence…

Line 212: This implies better quality of life for those blind or SVI before cataract surgery.

Use either 'severely visually impaired' or 'with SVI'

Line 224: Unlike other studies, we did not find age or presence of a complication as a risk factor for poor outcomes

Replace ‘as’ with ‘to be’

Line 226: No devastating complications like expulsive choroidal haemorrhage or endophthalmitis were report, though the number of surgeries was less.

Reported, not ‘report’

Line 236: Patients with ocular comorbidities, can then be counselled with a clear explanation on the outcomes of surgery.

Delete comma

Line 239: However, when we compared those available and those lost to follow up, we found that they differed in terms of age group, operating surgeon, and presence of ocular comorbidity.

Suggest rephrase… there was a statistically significant difference between the two groups only in terms of age group, operating surgeon, and presence of ocular comorbidity.

Line 241: Those available for follow up had higher prevalence of ocular comorbidity.

‘had a higher’

Line 242: PRECOG study also showed good correlation between early outcome (3 or fewer days) and 40 days or more post operatively.

Delete ‘also’ (you have it in the next sentence too)

Line 245: Hence, the results obtained from those available for follow up can underestimate the good outcomes.

‘may’, instead of ‘can’

Conclusion: Rephrase to link the thoughts/clarify your recommendations. Eg. you might suggest, after your sentence re well trained staff, that the planned residency program is a move in such a direction. Whether regular monitoring would improve the catarat outcomes in patients with ocular comorbidities is doubtful- in this case, screening and prevention (where possible) of primary disease may be more relevant.

Tables:

Table 1: Please check formatting of table. ? no line between ‘HIV seropositive’ and ‘none’...?

Table 3: Since you are comparing between those whose follow up was available vs not, the percentages should be by column (eg gender: Female 222/285 = 77.9%). Likewise for table 4 (good vs poor outcome)

Reviewer #2: While commending the efforts of the authors , i will wish to state that this manuscript has significant flaws.

First is the title. A change ought to be considered to reflect the fact that the "risk factor" considered was for poor outcome. The title as it is "Cataract Surgery Visual Outcomes and Associated Risk Factors in Liberia" does not reflect this point.

The authors only considered visual acuity using distant snellen acuity and there was no information on near vision. It would have been of benefit to give an idea of the post operative refractive outcome ie myopia, hyperopia or astigmatism within a specified range of diopters.

A significant concern is a lack of description of the process. There is need to inform the readers of what process the patient has to go through. A step wise description of the process from the point of patient contact and diagnosis, through the preoperative workup and surgical technique should be included in the manuscript.

What was the basis for selection to perform MSICs or Phacoemulsification?

No information on biometry was given ie was biometry done and at what point?

There is no information on if there was a patient counseling, and by whom.

Was there a patient participation in decision to surgery? Patient consent was not mentioned, let alone discussed.

Can an explanation for the high number of failure to attend 4 to11 week follow up clinic visit be given.

There are other significant points requiring attention in the attached manuscript. This has been included as comments in the manuscript. Click on the comments icon at each point to view the question.

Reviewer #3: Why were most patients operated using MSICS and not phacoemulsification?

Only a total of 86.8% received intracameral antibiotics, what informed this decision and did this information affect any of the measured outcomes?

6. PLOS authors have the option to publish the peer review history of their article (what does this mean?). If published, this will include your full peer review and any attached files.

Reviewer #1: Yes: Evelyn Tai

Reviewer #2: Yes: OGUGUA, N. OKONKWO

Reviewer #3: No

---

## [Author Response · Author response to Decision Letter 0]

11 Apr 2020

Response to Editor comments

1. Should not the title be more reflective if it was: Factors associated with Visual Outcomes after Cataract Surgery: A cross-sectional or retrospective study in Liberia 

Our response: Thanks for this suggestion. Title has been modified accordingly. (Page1; Line 1-2)

2. Abstract: The objective seems to be good as a Title. 

Our response: Thanks for this feedback. 

3. Conclusion: Conclusions do not support the results. How do we know if these are similar to the WHO outcomes? Either it could be said “very high or give the data from WHO”.

Our response: Thanks for this feedback. The conclusion is modified accordingly. It says The cataract surgical outcomes in Liberia were good; with ocular comorbidities as the only risk factor.” (Page2; Line 20-21)

4. Results: Lines 119-122 should go under Materials and Methods. Also, same information is repeated twice. 

Our response: Thanks for the suggestion. However as the lines 119-122 are part of results, we would request to retain it in results. 

5. Table 2: While UCVA is better than 6/12 for 44 for UCVA 115 (20%), did they need any surgery? If they were referred in future, could have been saved resources and extend the facilities who need urgent treatment. Please explain the rationale behind it. 

Our response: Thanks for the comment. We agree with the editor that these group may have not needed surgery. However, as per our protocol, we leave the decision of surgery to patient and suggest them to undergo surgery if there is any visual disability. It’s well known that individual disability level cannot be confidently predicted from his / her visual acuity alone. 

Enclosed related reference: https://www.ncbi.nlm.nih.gov/books/NBK207559/

6. Table is very difficult to understand its meaning. How the outcome sustains in different VA categories would be appropriate. For example, of 115 (20%) had VA 6/12 at baseline compared to 82.5% after week 4. This is a fascinating improvement. 

Our response: Outcome is given as per WHO categories which makes it simple to understand.

7. Then, 289 (50.4%) had worse than 6/60 at baseline compared to 5.2% at week 4. This means, approximately 289-23= 266 improved their vision from 6/60, i.e. 92% got improvement. Is it correct? In case of 6/12 or better, 364-115 = 249 i.e., 68% got improvement. Worse the VA, more the improvement after surgery. Is it not the message? 

Our response: Thanks for the comment. However unfortunately, this is not the correct way of interpreting. The table signifies only change in categories. It’s likely that the 115 who were 6/12 or better remained in same category i.e. 100% improvement. Post-operative 6/12 or better means that are patients in lower categories pre-operatively and have moved to 6/12 or better post-operatively. 

8. Table 2: Please remove the % to fit and make the Table better. 

Our response: We would like to keep the ‘n’ and % as this gives better idea of the % who have improved /post-operatively. However, if you insist, we can remove it. 

9. Surgeon category: Did Faculty 2 performed worse? Is there any explanation of this significant finding? 

Our response: There was no difference in outcomes of the surgeons as shown in table 4. The p values for surgeon category in 0.9. 

10. Please combine Table 4 and 5 together. When the univariate analysis shows non-significant association, no need to go for multi-variate adjustment. In fact, all are non-significant except the last one which could be presented in the text. 

Our response: Table 4 has the ‘n’ in each group as well as uses the chi-square test. Table 5 is univariable and multivariable analysis. Both tables have different information and we would request to leave them for better understanding of readers. 

For multivariable analysis, selection does not depend on the results of univariable analysis. If one thinks that the independent variable will have an effect of dependent variable, the variable can be used in both analysis. We have accordingly considered all variable for univariable and multivariable analysis. For more details on how to select variable for univariable and multivariable analysis, please refer the following: http://www.biostathandbook.com/multiplelogistic.html

11. Conclusion: “In conclusion, this is the first study to report on the outcomes of cataract surgeries in Liberia. This is not a conclusion. It could be said “this first study conducted in Liberia reported……………….”. 

Permanent infrastructure, uniform protocols, and well-trained staff in all cadres can improve uptake of cataract surgeries in future. It is not clear with which results this conclusion is linked. 

Regular monitoring and follow-up will improve the outcomes further. 

A Residency program has been started in Liberia recently and this article is one of the first articles from Liberia and outcomes of residents will be compared and reported in future. Is it a conclusion or strength? Alternatively, are we interest to see what we have instead of potential outcome?

I would consider the conclusion should be better reflected of the findings. 

Our response: Thanks for your valuable inputs. The text is modified as: ‘This is the first study to report on the outcomes of cataract surgeries in Liberia. We believe that there are multiple factors which might have played a role - availability of full time well-trained surgeons, accurate biometry, availability of equipment and consumables, and well trained staff. However, further studies are needed to support this hypothesis. A Residency program has been started in Liberia recently and this article is one of the first articles from Liberia and outcomes of residents will be compared and reported in future.’ 

‘In conclusion, overall the outcomes of cataract surgery in Liberia was good as compared to many studies done in Africa. Apart from this, the complications rates were also comparable to WHO standards and only risk factor for poor outcome was presence of ocular comorbidities.’ (Page 16; Line 10-19)

Response to reviewer 1

1. This study presents the cataract surgery outcomes of the Liberia Eye Health Initiative by the LVPEI and highlights risk factors for poor outcome. The study is simple and the results clearly presented, but some grammar editing and proofreading is required to facilitate readability. Recalculation of some figures in the tables would also be helpful. I have included some suggestions below.

Our response: Thanks for your constructive criticism. 

2. Line 28: ‘Since July 2017, LV’ Putting this at end of sentence would improve flow

Our response: Thanks for this comment. Suggestion incorporated. It reads as ‘LV Prasad Eye Institute (LVPEI), Hyderabad, started providing eye care in Liberia since July 2017.’ (Page2; Line 4-5)

3. Line 44: recommendation, not ‘recommended’

Our response: Thanks for comment. However, based on editor comments the conclusion is changed and it reads as ‘The cataract surgical outcomes in Liberia were good; with ocular comorbidities as the only risk factor.’ (Page2; Line 20-21)

4. Line 51: of, not ‘for’

Our response: Thanks for pointing this out. Suggestion incorporated. It reads as “In Sub-Saharan Africa 35-45% of blindness and 25-35% of VI is caused by cataract” (Page3; Line 5)

5. Line 54: from blindness of which 50% is due to cataract. Add comma after blindness 

Our response: Thanks for the suggestion. This is done. (Page3; Line 8)

6. Line 54-56: In terms of human resources, there is only one ophthalmologist for one million population on an average in the sub-Saharan region. Rephrase: In terms of human resources, the sub-Saharan region has on average only one ophthalmologist per one million population.

Our response: Thanks for the suggestion. This is rephrased. It reads as ‘In terms of human resources, the sub-Saharan region has on average only one ophthalmologist per one million population’ (Page3; Line 8-9)

7. Line 56: This holds true even for Liberia [4] Add full stop at end of sentence. 

Our response: This is done and a full stop has been added. (Page3; Line 10)

8. Line 63-65 : The L V Prasad Eye Institute’s (LVPEI) pyramidal model of eye care delivery has a Centre of Excellence (CoE) at the top catering to a population of 50 million population with Tertiary Centres (TC) at the next level, each for 5 million population.[10] add commas after 'top' and '50 million' to improve clarity

Our response: this is done and the sentence now reads as “..delivery has a Centre of Excellence (CoE) at the top, catering to a population of 50 million population,…” (Page3; Line 18)

9. Line 80: The Liberia Eye Centre (LEC) was formally inaugurated on 24 July 2017 in Monrovia, to provide comprehensive eye care for people of Liberia. 

10. Our response: Thanks for pointing this out. The comma is deleted. The sentence now reads as “The Liberia Eye Centre (LEC) was formally inaugurated on 24 July 2017 in Monrovia to provide comprehensive eye care for people of Liberia” (Page4; Line 9)

11. Line 81-82: Since 2018, LVPEI also started the ophthalmology residency training program in Liberia. Suggest delete this line: irrelevant and disrupts the flow (unless this study involves this program?)

Our response: While point is well taken, the idea of introducing the residency program is to build a case for resident outcomes for our next paper which is on their outcomes. Hence we would request to retain it. 

12. Line 82-84: The purpose of this study is to evaluate outcomes of cataract surgery performed, at the LEC as well as associated risk factors for poor outcomes. Delete comma

Our response: Thanks for pointing this out. The comma is deleted. Sentence reads as ‘The purpose of this study is to evaluate outcomes of cataract surgery performed at the LEC as well as associated risk factors for poor outcomes. (Page4; Line 12)

13. Line 132: While 562 (98.1%) eyes underwent PCIOL, 11 (1.9 %) eyes were either left aphakic or had an ACIOL. Rephrase; try not to start sentence with 'while'

Our response: Thanks for pointing this out. Sentence changed and it reads as “Five hundred and sixty two (98.1%) eyes underwent PCIOL, 11 (1.9 %) eyes were either left aphakic or had an ACIOL” (Page 7; Line 6-7)

14. Line 133-136: Major intraoperative complications such as posterior capsular rest and zonular dehiscence were noted in 3.3% eyes. Other complications (0.9%) such as Descemet’s membrane detachment (1eye), iridodialysis (2 eyes) and would leak (2 eyes) were also noted. 3.3% 'of' eyes. Note spacing between ‘1’ and ‘eye’. Generally, when quoting figures below ten, they should be spelt in full (eg. one eye)

Our response: Thanks for pointing this out. Suggestion incorporated. Sentence reads as ‘Major intraoperative complications such as posterior capsular rest and zonular dehiscence were noted in 3.3% of eyes. Other complications (0.9%) such as Descemet’s membrane detachment (one eye), iridodialysis (two eyes) and would leak (two eyes) were also noted.’ (Page 7; Line 7-10)

15. Line 138-139: Table 2 shows the preoperative and post-operative (uncorrected and best corrected) visual acuity first day; 1 to 3 weeks and 4-11 weeks follow up. …visual acuity ‘on the’ first day

Our response: Thanks for your suggestion. It reads as ‘Table 2 shows the preoperative and post-operative (uncorrected and best corrected) visual acuity on the first day; 1 to 3 weeks and 4-11 weeks follow up.’ (Page 8; Line 2)

16. Line 162: Those with poor outcomes were older (p=0.04); had higher intraoperative complications (p<0.001) as well as associated ocular comorbidity (<0.001); and had no IOL and AC IOL (p=0.001). It would be clearer to say ‘and had either an ACIOL or were left aphakic’

Our response: Suggestion incorporated. It reads as ‘Those with poor outcomes were older (p=0.04); had higher intraoperative complications (p<0.001) as well as associated ocular comorbidity (<0.001); and had either an AC IOL or were left aphakic (p=0.001).’ (Page 10; Line 6)

17. Line 183: We reported here the outcomes of cataract surgeries performed by in a new facility built in Liberia. Replace with 'present'

Our response: Suggestion incorporated. It reads as ‘We reported here the outcomes of cataract surgeries performed by in a present facility built in Liberia.’ (Page 13; Line 7)

18. Line 185: ‘We started this new facility in July 2017, and installed the Electronic Medical Records (EMR) system to collect and store patient data on a daily basis. This is a good source for monitoring the outcomes of cataract surgery.’ Comment: This part is not relevant in discussion. Please delete. Any discussion of methods eg EMR should be in Methods.

Our response: Comments well taken, however, as EMR is relevant for prospective data collection, we feel its relevant to discuss its importance. It has been mentioned in methods as well as discussion. If all hospitals in Africa get EMR, routine monitoring of cataract outcome would become easy. 

19. Line 189: The only report from Liberia by Frucht-Pery and Feldman, on cataract surgical outcomes in patients with leprosy, showed that a visual acuity of 20/200 or better was achieved in 65% of 191 patients. Add comma after Liberia

Suggestion incorporated. It reads as ‘The only report from Liberia, by Frucht-Pery and Feldman, on cataract surgical outcomes in patients with leprosy, showed that a visual acuity of 20/200 or better was achieved in 65% of patients’ (Page 13; Line 13)

20. Line 191: However, this study published more than 25 years ago, included only 43 eyes and the surgical technique used was intracapsular or extracapsular cataract extraction. -grammar: ‘…this study, which was published ...’

Suggestion incorporated. It reads as ‘However, this study was published more than 25 years ago, included only 43 eyes and the surgical technique used was intracapsular or extracapsular cataract extraction’ (Page 13; ;Line 15)

21. Line 198-202: There are very few reports on outcomes of cataract surgery from Africa, and outcomes reported by these are not encouraging, except a few centres. One of the reasons for this was the availability of well-trained surgeons performing high volume work in these centres. However, outcomes were better than many other studies from Africa, including PRECOG study. This could be due to availability of well-trained surgeons, accurate biometry in all cases, availability of equipment and consumables, and well trained staff. Please rephrase for clarity and readability. For example, by 'one of the reasons for this', it is unclear if you mean the reason for the poor outcome or the exceptions. and when you say' outcomes were better than many other studies', to which outcomes do you refer?

Our response: Comment well taken and changes done. The sentence now reads as “One of the reasons for improved outcome may be the -availability of well-trained surgeons performing high volume work in these centres.” (Page 14; Line 9-11)

22. Line 208: The outcomes were also similar to the recommended rates of WHO for BCVA (BCVA of 90% having 6/18 or better), but were less than what is recommended for UCVA. Rephrase sentence…

Our response: Comment well taken and changes done. The sentence now reads as “WHO has recommended a good visual outcome as 90% having BCVA of 6/18 or better and 80% having UCVA of 6/18 or better. The outcomes were also similar to the recommended rates of WHO for BCVA, but were less than what is recommended for UCVA. (Page 14; Line 17-19)

23. Line 212: This implies better quality of life for those blind or SVI before cataract surgery. Use either 'severely visually impaired' or 'with SVI'

Our response: Comment well taken and changes done. The sentence now reads as ‘This implies better quality of life for those blind or with SVI before cataract surgery.’ (Page 14; Line 23)

24. Line 224: Unlike other studies, we did not find age or presence of a complication as a risk factor for poor outcomes. Replace ‘as’ with ‘to be’

Our response: Comment well taken and changes done. The sentence now reads as ‘Unlike other studies, we did not find age or presence of a complication to be risk factor for poor outcomes’ (Page 15; Line 11)

25. Line 226: No devastating complications like expulsive choroidal haemorrhage or endophthalmitis were report, though the number of surgeries was less. Reported, not ‘report’

Our response: Comment well taken and changes done. The sentence now reads as ‘No devastating complications like expulsive choroidal haemorrhage or endophthalmitis were reported, though the number of surgeries was less.’ (Page 15; Line 13)

26. Line 236: Patients with ocular comorbidities, can then be counselled with a clear explanation on the outcomes of surgery. Delete comma 

Our response: Comment well taken and changes done. The sentence now reads as ‘Patients with ocular comorbidities can then be counselled with a clear explanation on the outcomes of surgery.’ (Page 15; Line 22)

27. Line 239: However, when we compared those available and those lost to follow up, we found that they differed in terms of age group, operating surgeon, and presence of ocular comorbidity. Suggest rephrase… there was a statistically significant difference between the two groups only in terms of age group, operating surgeon, and presence of ocular comorbidity.

Our response: Comment well taken and changes done The sentence now is rephrased as “However, on comparing those available and those lost to follow up, there was a statistically significant difference between the two groups only in terms of age group, operating surgeon, and presence of ocular comorbidity.” (Page 15; Line 25 and Page 16; Line 1-2)

28. Line 241: Those available for follow up had higher prevalence of ocular comorbidity. ‘had a higher’

Our response: Comment well taken and changes done. The sentence now reads as ‘Those available for follow up had a higher prevalence of ocular comorbidity’ (Page 16; Line 2)

29. Line 242: PRECOG study also showed good correlation between early outcome (3 or fewer days) and 40 days or more post operatively. Delete ‘also’ (you have it in the next sentence too)

Our response: Comment well taken and changes done. The sentence now reads as ‘PRECOG study showed good correlation between early outcome (3 or fewer days) and 40 days or more post operatively.’ (Page 16; Line 4)

30. Line 245: Hence, the results obtained from those available for follow up can underestimate the good outcomes. ‘may’, instead of ‘can’

Our response: Comment well taken and changes done. The sentence now reads as ‘Hence, the results obtained from those available for follow up may underestimate the good outcomes.’ (Page 16; Line 7)

31. Conclusion: Rephrase to link the thoughts/clarify your recommendations. Eg. you might suggest, after your sentence re well trained staff, that the planned residency program is a move in such a direction. Whether regular monitoring would improve the catarat outcomes in patients with ocular comorbidities is doubtful- in this case, screening and prevention (where possible) of primary disease may be more relevant.

Our response: Comment well taken, however based on editor comment, the conclusion has been changed. It reads as “In conclusion, overall the outcomes of cataract surgery in Liberia was good as compared to many studies done in Africa. Apart from this, the complications rates were also comparable to WHO standards and only risk factor for poor outcome was presence of ocular comorbidities. (Page 16; Line 16-19)

32. Tables:Table 1: Please check formatting of table. ? no line between ‘HIV seropositive’ and ‘none’...? 

Our response: Comment well taken and a line has been added

33. Table 3: Since you are comparing between those whose follow up was available vs not, the percentages should be by column (eg gender: Female 222/285 = 77.9%). Likewise for table 4 (good vs poor outcome)

Our response: Comment well taken and is done for table 3 and 4

Response to reviewer 2

1. While commending the efforts of the authors , i will wish to state that this manuscript has significant flaws. First is the title. A change ought to be considered to reflect the fact that the "risk factor" considered was for poor outcome. The title as it is "Cataract Surgery Visual Outcomes and Associated Risk Factors in Liberia" does not reflect this point.

Our response: Thanks for your constructive criticism. We will try our best to address all the issues raised. As far as title is concerned, based on editor suggestion, the title has been modified. Please see response to editor comment 1. 

2. The authors only considered visual acuity using distant snellen acuity and there was no information on near vision. It would have been of benefit to give an idea of the post operative refractive outcome ie myopia, hyperopia or astigmatism within a specified range of diopters. 

Our response: Thanks for your valuable suggestion. Pre and post-operative refractive error change as well as presbyopia are part of a different manuscript. 

3. A significant concern is a lack of description of the process. There is need to inform the readers of what process the patient has to go through. A step wise description of the process from the point of patient contact and diagnosis, through the preoperative workup and surgical technique should be included in the manuscript. 

Our response: As suggested, a detail description is included. It reads as ‘The protocols were similar to described in our previous publication.[11] In brief, the patients underwent comprehensive eye examination, which included detailed history; uncorrected visual acuity (UCVA) and best corrected visual acuity (BCVA); intraocular pressure measurement with Goldmann applanation tonometer; slit lamp examination; dilated lens examination to assess the lens status; and stereoscopic fundus examination with +78/90 Dioptre lens as well as indirect ophthalmoscope. In case there was no view of retina, a B-scan ultrasound was done to rule out any posterior segment pathology. Pre-existing ocular comorbidities were grouped as corneal pathologies, retinal disease, glaucoma and others (non-glaucomatous optic nerve pathologies and uveitis). The systemic comorbidities included hypertension (HT), diabetes mellitus (DM), and HIV seropositive patients. When patient was advised surgery, protocol similar to L V Prasad Eye Institute protocols were followed.[12, 13] In brief, when patient was advised surgery, a designated counsellor did the counselling and explained the type of surgeries as well as associated risk and benefits. A day prior to surgery, intraocular lens (IOL) power calculation was done by measuring keratometry and A Scan biometry. Informed consent from patient and attendant was also taken along with routine blood pressure and blood sugar measurement and a physician fitness a day prior to surgery. On day of surgery, prior to entering operating room, patient dress was changed and eye were dilated with plain tropicamide (0.8% w/v) eye drop. Local anaesthesia given with 2% Xylocaine. After local anaesthesia, patient was shifted to operating room and under all aseptic precautions, eye was cleaned with betadine and draped. The surgeon decided on the surgical technique – either a phacoemulsification or manual small incision cataract surgery (MSICS) with or without intraocular lens (IOL) implantation. MSICS was performed by standard Blumenthal technique.[14] For phacoemulsification, a 5.5 mm scleral would was constructed and a routine phacoemulsification was performed. The choice of procedure was left to surgeon discretion. Anterior chamber (AC) and posterior chamber (PC) IOLs made up of polymethyl methacrylate (PMMA) were used. The intraoperative complications (posterior capsule rupture, zonular dehiscence etc) were noted. Posterior capsular rent or zonular dehiscence was managed by automated vitrectomy and placement of IOL was based on the available support of anterior and / or posterior capsule.

All the surgeries were performed by three surgeons as well as other visiting faculty. All three surgeons had experience of performing more than 1,500 cataract surgery. Post-operatively, patient was prescribed topical steroids for 4 weeks in tapering doses and topical antibiotics for a week. (Page 4; Line 22-25; Page 5; Line 1-25 and Page 6; Line 1-4 )

4. What was the basis for selection to perform MSICs or Phacoemulsification? 

Our response: As described in the manuscript, the choice of procedure was left to surgeon discretion. (See response to comment 3)

5. No information on biometry was given ie was biometry done and at what point?

Our response: Thanks for the comment. This is added to the manuscript and the sentence reads as ‘A day prior to surgery, intraocular lens (IOL) power calculation was done by measuring keratometry and A Scan biometry.’ (See response to comment 3)

6. There is no information on if there was a patient counseling, and by whom. 

Our response: Thanks for the comment. This is added to the manuscript and it reads as ‘In brief, when patient was advised surgery, a designated counsellor did the counselling and explained the type of surgeries as well as associated risk and benefits.’ (See response to comment 3)

7. Was there a patient participation in decision to surgery? Patient consent was not mentioned, let alone discussed.

Our response: Thanks for the comment. This is added to the manuscript and it reads as ‘Informed consent from patient and attendant was also taken along with routine blood pressure and blood sugar measurement and a physician fitness a day prior to surgery.’ (See response to comment 3)

8. Can an explanation for the high number of failure to attend 4 to11 week follow up clinic visit be given

Our response: The follow-up rate was nearly 77% which is much better than most of the studies done in Africa. We are separately collecting data on barriers which is part of different manuscript.

9. There are other significant points requiring attention in the attached manuscript. This has been included as comments in the manuscript. Click on the comments icon at each point to view the question.

Our response: Thanks for this information.

10. In my opinion , this title does not read well and is not clear. It may be preferable to say "Outcome of Cataract Surgery and Risk Factors for Poor Visual Outcome Amongst Liberians / or In Liberians. I am very aware of your short title.

Our response: Thanks for this suggestion. As far as title is concerned, based on editor suggestion, the title has been modified. Please see response to editor comment 1. The short title now is changed to “Visual outcomes of cataract surgery in Liberians” (Page1; Line 21)

11. Risk factors for what? If it is for poor visual outcome, then please say so and add it to the sentence.

Our response: Thanks for this suggestion. It is changed to ‘To report the initial outcomes and associated risk factors for poor outcome of cataract surgery performed in Liberia’ (Page2; Line 2-3)

12. In my mind this should read "Majority of the patients underwent MSICS, while a few had phacoemulsification" . This is because only 5.4% of the eyes had a phaco. according to your results, which is far fewer than the number than had MSICS.

Our response: Thanks for this suggestion. This has already been mentioned in results. 

13. Was this finding also at 4 -11 weeks ? 

Our response: Yes, this was at 4-11 weeks also

14. What factors influenced the decision on choice of technique, IOL use ?

Our response: It was the surgeon who decided on the surgical technique. Anterior chamber (AC) and posterior chamber (PC) IOLs made up of polymethyl methacrylate (PMMA) were used. (See response to comment 3)

15. Please provide details of the experience and expertise of the three surgeons and the visiting faculty. Were cases done my ophthalmology residents +/- with supervision

Our response: The three surgeons have performed more than 1500 cataract surgeries in India. No surgeries were performed by the residents. (See response to comment 3)

16. Also provide more details of the entire pre operative process. Was a biometry done and at what point? 

Our response: (See response to comment 3)

17. Was there a need for B scan ultrasonography in some cases? 

Our response: Yes, five patients underwent ultrasonography

18. What of blood works etc? 

Our response: Yes, routine blood pressure measurement and blood sugar was checked and physician fitness was taken a day prior to surgery. (See response to comment 3)

19. Describe in detail the process each patient is expected to have gone through until he or she gets to the OR. I would also expect a description of the technique of MSICS including wound size etc. 

Our response: A day prior to surgery, intraocular lens (IOL) power calculation was done by measuring keratometry and A Scan biometry. Informed consent from patient and attendant was also taken along with routine blood pressure and blood sugar measurement and a physician fitness a day prior to surgery. On day of surgery, prior to entering operating room, patient dress was changed and eye were dilated with plain tropicamide (0.8% w/v) eye drop. Local anaesthesia given with 2% Xylocaine. After local anaesthesia, patient was shifted to operating room and under all aseptic precautions, eye was cleaned with betadine and draped. The surgeon decided on the surgical technique – either a phacoemulsification or manual small incision cataract surgery (MSICS) with or without intraocular lens (IOL) implantation. MSICS was performed by standard Blumenthal technique.[14] For phacoemulsification, a 5.5 mm scleral would was constructed and a routine phacoemulsification was performed. The choice of procedure was left to surgeon discretion. Anterior chamber (AC) and posterior chamber (PC) IOLs made up of polymethyl methacrylate (PMMA) were used. The intraoperative complications (posterior capsule rupture, zonular dehiscence etc) were noted. Posterior capsular rent or zonular dehiscence was managed by automated vitrectomy and placement of IOL was based on the available support of anterior and / or posterior capsule. Reference for the details of the tehnnique aand protocols has been added.

a. Thomas R, Kuriakose T, George R. Towards achieving small-incision cataract surgery 99.8% of the time. Indian J Ophthalmol. 2000 Jun;48(2):145-51. Review. PubMed PMID: 11116514.

b. https://www.lvpei.org/patient-care/surgeries-inpatient

c. https://www.lvpei.org/sub-speciality/cataract

(See response to comment 3)

20. Did all the surgeon employ the same technique and were there modifications. 

Our response: Yes: All surgeons followed the same technique (mentioned above). For MSICS, standard Blumenthal technique was used. For phacoemulsification, a 5.5 mm scleral would was constructed and a routine phacoemulsification was performed. (See response to comment 3)

21. What was done in cases of PC rent or zonular dialysis i.e how was this complication managed? Please give details. 

Our response: Posterior capsular rent or zonular dehiscence was managed by automated vitrectomy and placement of IOL was based on the available support of anterior and / or posterior capsule. (See response to comment 3)

22. At what point was patient consent for surgery taken? 

Our response: Informed consent from patient and attendant was also taken along with routine blood pressure and blood sugar measurement and a physician fitness a day prior to surgery. (See response to comment 3)

23. Was any form of preoperative counselling done? 

Our response: In brief, when patient was advised surgery, a designated counsellor did the counselling and explained the type of surgeries as well as associated risk and benefits. (See response to comment 3)

24. Did the patient participate in the decision of for surgery and what type of surgery to be done? 

Our response: In brief, when patient was advised surgery, a designated counsellor did the counselling and explained the type of surgeries as well as associated risk and benefits. (See response to comment 3)

25. There is no mention of a consent for surgery from the patient or relations. 

Our response: Informed consent from patient and attendant was also taken along with routine blood pressure and blood sugar measurement and a physician fitness a day prior to surgery. (See response to comment 3) 

26. What is the pre operative counselling like and at what stage is consent for surgery taken? 

Our response: In brief, when patient was advised surgery, a designated counsellor did the counselling and explained the type of surgeries as well as associated risk and benefits. (See response to comment 3)

27. can any explanation or discussion be given for the relatively higher amount / percentage or ratio of failure to attend follow up visits compared to attending follow up visits, amongst the "visiting faculty" . The ratio is very large and not seen in the other faculty.

Our response: This is a good observation. Visiting faculty were two local surgeons and two who came from India. It’s likely that as the visiting surgeon (their operating surgeon) was not available during the subsequent visit of the patient, the patients might have not turned up. 

28. poor outcome cannot be defined as BCVA of less than 6/12 . That means the 6/18 will fall into the category of poor vision. This truly is not the case.

Our response: This is as per WHO definitions. Also in most of the countries would have 6/12 vision mandatory for driving licence so that too was considered separately. 

29. Was any consideration given to the presence of posterior capsular opacity formation? Was YAG laser capsulotomy attempted in any of the eyes ?

Our response: As the follow up period was only 4-11 weeks, there was no YAG laser attempted. 

30. Please reconstruct this sentence. it is faulty and does not read well.

Our response: The sentence is reconstructed. It reads as ‘We reported here the outcomes of cataract surgeries performed in a present facility built in Liberia’ (Page 13; Line 7-8)

31. This is a repetition of information already given in the results section. It should do better if re phrased.

Our response: Sentence changed to ‘UCVA of 6/18 or better was seen in 63.5% patients and BCVA of 6/18 or better was seen in 88% patients at last follow up. With further cut-off in visual acuity value to 6/12 or better, UCVA of 6/12 or better was seen in 38.6% patients and BCVA of 6/12 or better was seen in 82.5% patient’. (Page 14; Line 3-6)

32. This statement is likely not the case .I recommend you visit the African Journal on Line (AJOL) to conduct your search on this topic. You will find a large amount of reports.

Our response: Thanks for the suggestion. New reference has been added and sentence changed to ‘There are very few reports on outcomes of cataract surgery from Africa, and outcomes reported by some of these are not encouraging’ (Page 14; Line 8-9)

33. These references are old . Over a decade old. Can there be more recent information. You may find this ref useful

Our response: Thanks for the suggestion. New reference added. 

Oderinlo, Olufemi & Hassan, Adekunle et al. (2016). Refractive aim and visual outcome after phacoemulsification: A 2-year review from a Tertiary Private Eye Hospital in Sub-Saharan Africa. Nigerian Journal of Clinical Practice. 20. 10.4103/1119-3077.183249.

34. Which centers are you referring to ? Please clarify .

Our response: These refer to centres in Africa. Sentence changed and reads as ‘One of the reasons for improved outcome in some centres may be the availability of well-trained surgeons performing high volume work in these centres.’ (Page 14; Line 9-11)

35. which outcomes is being referred to here? Is it outcome from this study ? Please state clearly and make your sentences clear and easy to understand.

Our response: Yes, we are referring to outcomes of our study. Sentence changed and it reads as ‘However, outcomes of this study was better than many other studies from Africa, including PRECOG study’. (Page 14; Line 11-12)

36. No mention was made about the training of the surgeons either in the methods section or hear. Faculty 3 appeared to perform far less surgery as well as the visiting faculty . Any comments on this ?

Our response: The three surgeons have performed more than 1500 cataract surgeries in India. It has been added in methods section. It reads as ‘All three surgeons had experience of performing more than 1,500 cataract surgery.’ The surgeries of faculty 3 is less as the surgeon joined at a later date. The visiting faculty were four in number and they came only for a week period. Hence, their numbers are less. (See response to comment 3)

37. No mention was made of biometry throughout the manuscript until now. How and when was this done. This has been previously mentioned in the methods section.

Our response: This has been previously mentioned in the methods section. It reads as ‘A day prior to surgery, intraocular lens (IOL) power calculation was done by measuring keratometry and A Scan biometry.’ (See response to comment 3)

38. Please reconstruct this sentence to provide clarity and meaning to it.

Our response: Its reconstructed. Reads as ‘This implies better quality of life for those blind or with SVI before cataract surgery.’ (page 14; Line 23)

39. This truly shows why a description of your process is essential, to understand what exactly the patient goes through from the point of initial patient contact in the clinic to after surgery care .

Our response: Thanks for the comments. It has been mentioned in methods section. (See response to comment 3)

40. A significant proportion of your patients did not complete the followup or dropped out. What will you do to reverse this trend?

Our response: The follow-up rate was nearly 77% which is much better than most of the studies done in Africa. We are separately collecting data on barriers which is part of different manuscript.

41. What outcomes of residents ? Please clarify.

Our response: Thanks for the comment. Sentence changed ‘A Residency program has been started in Liberia recently and this article is one of the first articles from Liberia and cataract surgery outcomes of residents will be compared and reported in future.’ (Page 16; Line 13-16)

Response to reviewer 3

1. Why were most patients operated using MSICS and not phacoemulsification?

Our response: As described in the manuscript, the choice of procedure was left to surgeon discretion. (See response to comment 3 of reviewer 2)

2. Only a total of 86.8% received intracameral antibiotics, what informed this decision and did this information affect any of the measured outcomes? 

Our response: This is a routine protocol of cataract surgery that is followed based on our previous experience of using intracameral antibiotics as a prophylaxis for prevention of endophthalmitis. However, in some cases, the surgeon missed to give it.

---

## [Editor Report · Decision Letter 1]

29 Apr 2020

Factors associated with Visual Outcomes after Cataract Surgery: A cross-sectional or retrospective study in Liberia

PONE-D-19-34050R1

Dear Professor Khanna,

We are pleased to inform you that your manuscript has been judged scientifically suitable for publication and will be formally accepted for publication once it complies with all outstanding technical requirements.

With kind regards,

Fakir M Amirul Islam, PhD

Academic Editor

PLOS ONE

Additional Editor Comments (optional):

Dear Authors,

Congratulations on your work. It is accepted. However, I would suggest to fix the formatting issues before it publishes. I have not liked the formatting of the Tables which do not look fantastic. For example,

Table 1: Number of Patients (percentage). If it gives 38 (6.6%) and so on. Does it not using the percentage twice? Either it should be removed from the title line or from the numbers. Removing from the numbers (inside of the tables) is preferred because it will give extra space to accommodate not for Table 1 but for other tables.

Table 2 is taking two rows due to keeping the percentage. It could be looked better. Still OK

Table 3: There is no title in the first two columns (characteristics and labels of characteristics). These two columns could be presented in one column bringing the labels of characteristics under the characteristics. This would give ample space to fit everything in Tables 3 & 4. I WOULD NOT KEEP THIS SILLY MISTAKES AND FORMATTING ISSUES LEFT FOR THE REVIEWERS/EDITOR. P values are mainly with two decimal places but one is with one decimal place. Sometimes, there are spaces between parenthesis and words and sometimes not. Should be very consistent throughout.

There are some footnotes in Table 3. However, the symbol choice seems random. *, plus sign (+), cross sign...........comes before these. These are formatting issue which is just a matter of last-minute work to fix after a long time work.
---

## [Editor Report · Acceptance letter]

5 May 2020

PONE-D-19-34050R1 

Factors associated with Visual Outcomes after Cataract Surgery: A cross-sectional or retrospective study in Liberia 

Dear Dr. Khanna:

I am pleased to inform you that your manuscript has been deemed suitable for publication in PLOS ONE. Congratulations! Your manuscript is now with our production department. 

With kind regards,

on behalf of

Dr Fakir M Amirul Islam 

Academic Editor

PLOS ONE